# Therapeutic Potential of Marine Peptides in Prostate Cancer: Mechanistic Insights

**DOI:** 10.3390/md20080466

**Published:** 2022-07-22

**Authors:** Salman Ahmed, Waqas Alam, Philippe Jeandet, Michael Aschner, Khalaf F. Alsharif, Luciano Saso, Haroon Khan

**Affiliations:** 1Department of Pharmacognosy, Faculty of Pharmacy and Pharmaceutical Sciences, University of Karachi, Karachi 75270, Pakistan; salmanahmed@uok.edu.pk; 2Department of Pharmacy, Abdul Wali Khan University, Mardan 23200, Pakistan; waqasalamyousafzai@gmail.com; 3Research Unit “Induced Resistance and Plant Bioprotection”, Department of Biology and Biochemistry, Faculty of Sciences, University of Reims, EA 4707-USC INRAe 1488, SFR Condorcet FR CNRS 3417, P.O. Box 1039, CEDEX 02, 51687 Reims, France; philippe.jeandet@univ-reims.fr; 4Department of Molecular Pharmacology, Albert Einstein College of Medicine, Bronx, NY 10461, USA; michael.aschner@einsteinmed.org; 5Department of Clinical Laboratory, College of Applied Medical Science, Taif University, P.O. Box 11099, Taif 21944, Saudi Arabia; alsharif@tu.edu.sa; 6Department of Physiology and Pharmacology, “Vittorio Erspamer” Sapienza University, 00185 Rome, Italy; luciano.saso@uniroma1.it

**Keywords:** marine peptides, apoptosis, antimetastatic, antimitotic, antiangiogenic, cell cycle arrest

## Abstract

Prostate cancer (PCa) is the leading cause of cancer death in men, and its treatment is commonly associated with severe adverse effects. Thus, new treatment modalities are required. In this context, natural compounds have been widely explored for their anti-PCa properties. Aquatic organisms contain numerous potential medications. Anticancer peptides are less toxic to normal cells and provide an efficacious treatment approach via multiple mechanisms, including altered cell viability, apoptosis, cell migration/invasion, suppression of angiogenesis and microtubule balance disturbances. This review sheds light on marine peptides as efficacious and safe therapeutic agents for PCa.

## 1. Introduction

Prostate cancer (PCa) is one of the most often diagnosed cancers worldwide. It is the second leading source of cancer-related mortality in males, trailing behind only lung cancer, based on GLOBOCAN 2020 estimates [1]. Radiation and surgical procedures are used to treat this disease when it first appears and is localized. Despite a considerable increase in disease-free life following first surgical or radiation therapy, the illness recurs in more than 30% of patients. Androgen deprivation treatment (ADT) is the most often alternative for PCa treatment, because of the tumor’s requirement for male hormones for progression. This treatment is focused on pharmacological castration achieved via GnRH agonists alone or in conjunction with anti-androgens. However, despite an excellent initial response, most patients relapse within 2–3 years, and the tumor advances. Chemotherapeutic drugs, such as docetaxel, cabazitaxel, doxorubicin, abiraterone and enzalutamide, provide a few months of progression-free survival with highly toxic effects. Therefore, the development and refinement of unique anticancer drugs with minimal adverse side effects are required [2,3].

Natural products provide benefits over synthetic compounds because of their broader variety of targets, larger structural diversity and low toxicity against PCa [4,5]. Many bioactive substances found in the marine ecosystem have potential applications in the treatment of human diseases including cancer. A large number of novel sea-based biological compounds have been derived from corals, sponges, tunicates, bacteria, fungi, micro- and macroalgae, and other marine micro- and macro-organisms. Treatments have been impacted by marine-based medicines, with anticancer treatments made available with several marine chemical compounds. Indeed, clinical approval has been granted for several such treatments, including belantamabmafodotin, cytarabine, enfortumabvedotin, brentuximab vedotin, eribulin mesylate, lurbinectedin, fludarabine phosphate (prodrug of ara-A), nelarabine (prodrug of ara-G), polatuzumabvedotin, trabectedin, vidarabine, and plitidepsin [6,7,8,9]. Improvement in the physiological state of cancer patients depends on the introduction of natural strategies for clinical treatment. Due to their small size, ease of synthesis, capacity to cross cell membranes, low drug–drug interactions, precision targeting, and reduced side responses, marine peptides have also spurred attention in the development of anticancer medicines. The downsides of anticancer peptides include their short half-life, low bioavailability, poor pharmacokinetics, and protease sensitivity [10,11,12]. A schematic representation of the pathophysiology of prostate cancer is shown in Figure 1.

Presently, more than 60% of clinically accessible anticancer medications have been derived from natural sources. Peptides are categorized depending on their ability to induce antioxidative mechanisms, cytotoxicity, apoptosis, inhibit cell proliferation, migration, and angiogenesis, microtubule-destabilization, or undiscovered mechanisms in diverse malignant cell lines, relating to clinical research for cancer treatment evaluation [12,13]. Anticancer peptides from algae, ascidians, bacteria, fungi, cyanobacteria, mollusks, sponges, and protein hydrolysates from clam, coral and fish, have been isolated. Linear and cyclic peptides are the two types of marine peptides. A straight-chain of amino acids linked together by amide bonds forms a linear peptide (Figure 2) [7]. HTI-286 [14], SHP [15], SIO [16] (tripeptides); Microsporin A [17], SCH-P9 and SCH-P10 [18] (tetrapeptides); dolastatin 10 [19], ILYMP [20] (pentapeptides); AAP-H (oligopeptide) [21] are reported from mollusk, sponges, fish, clam, sea anemone and marine derived fungus. Cyclic tetra, penta, hepta, octa, dodecapeptides and depsipetides are claimed to have anti-PCa effects. Cyclic depsipeptides have more complex structures, where successive ester linkages replace more amide bonds due to the presence of hydroxy acids in the peptide structure [7]. Aurilide B [22], lagunamide C [23], cryptophycin-52 [24] and coibamide A [25] are obtained from cyanobacteria. Geodiamolides D–F [26], homophymines A–E [27], jaspamide [28,29] and neamphamides B–D [30] from sponges are cyclic depsipeptides with anti- cancer effects. Kahalalide F from (mollusk) [31]; tamandarins A,B (ascidia) [32,33] sansalvamide A (fungus) [34] are other reported anticancer cyclic depsipeptides. Furthermore, microsporin A (cyclic tetrapeptide) [17], zygosporamide (cyclic pentadepsipeptide) [35] rolloamide A [36] and trunkamide A (a cyclic heptapeptide) [12,37], patellamides B and F (cyclic octapeptides) [38] and laxaphycin B (a cyclic dodecapeptide) [39] have been isolated from ascidia, cyanobacteria, sponge and marine derived fungus. Protein hydrolysates, the complex mixtures of oligopeptides and free amino acids have antioxidant, antiproliferative, antihypertensive, and antibacterial properties [40,41,42]. Protein hydrolysates obtain from mollusk [43], fish [15,16,44,45] and clam [18,20] have anti-PCa properties.

Several physiological and molecular mechanisms are used by marine anticancer peptides, including DNA repair, apoptosis induction, cell-cycle regulation, angiogenesis inhibition, migration, invasion, and metastasis suppression (Table 1) [8,46]. This review emphasizes marine anti-PCa peptides and their significance in the development of novel anticancer therapeutics.

## 2. Mechanistic Insights 

### 2.1. Apoptosis

One of the most critical mechanisms of cell death is apoptosis, and its failure is a severe barrier to cancer therapy. Cytochrome-c (cyt c) release leads to caspases activation and ensuing apoptosis [47]. *C*-phycocyanin from cyanobacteria [48] and AAP-H from sea anemone [21] have shown apoptotic efficacy by initiating cyt c release in DU145 and LNCaP cells. Caspases are the main executors of apoptosis, which are activated after proteolytic cleavage. Initiator caspases that include caspase-8, -9 and -10 initiate a regulated and programmed cell death cascade to trigger downstream caspases-3, -6, and -7 expression [49]. Tachyplesin, a cyclic peptide from horseshoe crab, triggers cyt c release, increasing caspases-3, -6, -7, -8, and -9 expression in TSU cells with IC_50_ of 75 μg mL^−1^ [50]. AAP-H from sea anemone has shown apoptotic efficacy by initiating cyt c release, enhancing caspases-3 and -9 activity in DU145 cells [21]. Cryptophycin-52 increases caspases-3 and -7 activities in DU145 and LNCaP cells [24]. The chromopeptide A from *Chromobacterium* sp. also increases caspase-3 activities and PARP cleavage in PC-3, DU145 and LNCaP cells [51]. Similarly, protein hydrolysates from clam, such as ILYMP, SCH-P9 and SCH-P10 [18,20] and MCH from mollusk have shown efficacy in DU145 and PC-3 [43]. SHP and SIO from fish increase caspase-3 activity in PC-3 and DU145 with IC_50_s of 15 and 1 mg mL^−1^, respectively [15,44,45]. 

Bcl-2 inhibition and BAX induction represents another method for initiating apoptosis [52]. *C*-phycocyanin induces apoptosis via caspases-3 and -9 activation, increasing BAX and decreasing Bcl2 and Bcl-xL in human prostate carcinoma DU145 and LNCaP cell lines with the IC_50_s in the range of 1-10 pM [48]. When DU145 cancer cells are treated with AAP-H oligopeptide, Bcl-2 is reduced, an effect related to the increased production of BAX, with IC_50_ of 2.298 mM [21]. Similarly, Sepia ink peptides SHP and SIO have been shown to induce apoptosis in PC-3 and DU145 by upregulating BAX and reducing Bcl-2 [15,44,45]. ILYMP initiates the phosphorylation of Bcl-2 and increases BAX in DU145 cells with IC_50_ of 11.25 mM [20]. Similarly, SCH-P9 and SCH-P10 from clam [18] and MCH from mollusk have shown the same behavior in DU145 and PC-3 [43].

PI3K/AKT pathways play a significant role in regulating cell cycle and survival. AKT inhibitors attenuate the degree of BAK, BAX, and BAD phosphorylation, cause cyt c release, and activate casp-9 [53]. Decreased PI3K/AKT and ErbB3 levels are involved in cell cycle arrest as well as BAX and BAK activation [54]. PI3K/AKT and ErbB3 deficiency in PC-3 and DU145 have been noted upon treatment with Kahalalide F [31,55]. Elisidepsin or Irvalec (a Kahalalide F synthetic derivative) have been shown to inhibit PI3K/AKT and deplete ErbB3 in PC-3 and DU145 cells [56,57]. Furthermore, elisidepsin causes cellular swelling, plasma membrane rupture, and loss of intracellular contents, as well as necrotic cell death in PC-3 and 22RV1 at IC_50_s of 0.6M and 0.3M, respectively [58]. p38 mitogen-activated protein kinases (MAPKs) and Jun *N*-terminal kinases (JNKs) are activated by microtubule inhibitors, suggesting this may represent a general stress response to microtubule dysfunction. Cyt c release is induced by JNK and p38 MAPK activation, which, in turn, triggers caspase cascades. Activation of JNK and ERK induces mitochondrial-related apoptosis via JNK signaling and S phase cell cycle arrest via ERK signaling [59,60]. Cryptophycin-52 induces apoptosis in DU145 and LNCaP cells via caspases-3, -7; JNK, p38 MAPK and ERK activation, increasing BAX and decreasing Bcl2 and Bcl-xL expression (Figure 3) [24]. 

Apoptosis stress involves mitochondrial outer membrane permeabilization via uncontrolled BH3 only proteins. BH-3 only proteins lead to oligomerization of BAK/BAX multimers. These BAK/BAX multimers within the outer membrane of the mitochondria form pores that allow cytochrome C release. Released cytochrome C interacts with Apaf-1 and pro-caspase-9 to form the apoptosome. Upon release, mitochondria-derived activator of caspase (SMAC) Cytochrome C and Omi activate apoptosome from procaspase-9 and cytochrome C. Caspases upon activation results in the cleavage of cellular proteins that leads to apoptosis. “Activation” is represented by blue arrows, whereas red T-bars show “inhibition”.

### 2.2. Antimitotic Effect

Antimitotic drugs function by stabilizing and destabilizing microtubule dynamics, as well as shifting the balance between tubulin polymerization and depolymerization. The majority of these drugs act via G2/M phase arrest [61]. Microtubules provide a variety of critical cellular activities, including chromosomal segregation, cell shape preservation, transport, motility, and organelle distribution. Microtubules, the key components of the mitotic spindle, play an important role in cell division. Microtubular dynamic disruption arrests the cell cycle at the metaphase–anaphase transition leading to cell death [62]. Hemiasterlin and its analogue HTI-286 depolymerize microtubules by disrupting microtubular dynamics in LNCaP, C4-2, PC-3, PC-3dR cell lines with IC_50_s in the range of 0.65–4.6 nM. The same effect has been noted in PC3-MM2, PC-3 and PC-3dR xenografts at 1–1.5 mg/kg i.v. [26,63]. Dolastatin 10 (IC_50_:0.5 nM) inhibits microtubule assembly in DU145 cells [19]. Analogous behavior has been observed for Diazonamide A in PC-3 cells with IC_50_ of 2.3 nM [64]. Cryptophycin-52 (LY355703), a synthetic cryptophycin, inhibits DU145 and LNCaP cell growth during mitosis by depolymerizing spindle microtubules and alters chromosomal organization [24]. By attaching to the microtubules, microtubule-stabilizing drugs promote microtubule polymerization and target the cytoskeleton and spindle apparatus of tumor cells, leading to mitotic interruption [62]. Aurilide B has been shown to cause microtubular destabilization in PC-3 and DU145 carcinoma cell lines with GI50 < 10 nM [22].

### 2.3. Antimetastatic Activity

Non-caspase proteases (elastase, trypsin and chymotrypsin) are critical regulators of PCa progression. The PCa metastatic cascade is characterized by a defined chain of steps, beginning with neoangiogenesis or lymphangiogenesis, culminating in the loss of tumor cell adhesion, local invasion of host stroma, and tumor cell escape into the vasculature or lymphatics, and eventually dissemination, extravasation, and colonization of specific metastatic sites. Proteases secrete angiogenic factors, cell adhesion molecules, breakdown basement membranes, induce epithelial–mesenchymal transition, participate in extravasation, and are necessary for metastatic site colonization. Several proteases are increased in tumor cells, and have specific roles in facilitating various phases of this cascade [65,66]. Trypsin plays a tumorigenic role in PCa and suppressing trypsin/mesotrypsin activity may provide a new PCa therapeutic strategy. PC-3 cells originating from a grade IV prostate cancer bone metastases exhibit an extremely significant overexpression of PRSS3/mesotrypsin [67]. LNCaP human prostate cells have shown upregulation of chymotrypsin-like proteasomal activity, suggesting the involvement of chymotrypsin in PCa [68]. Elastase increases PCa proliferation, migration, invasion and has been used as a therapeutic target [69]. Symplocamide A blocks chymotrypsin and trypsin with IC_50_s of 0.38 and 80.2 μM, respectively [70]. Kempopeptin A inhibits porcine pancreatic elastase (0.32 μM) and bovine pancreatic α-chymotrypsin (2.6 μM), whereas, Kempopeptin B only inhibits trypsin activity (8.4 μM) [71]. Bouillomides A and B inhibit elastase and chymotrypsin from porcine pancreas [72]. Molassamide, a depsipeptide from the cyanobacteria *Dichothrixutahensis, inhibits* porcine pancreatic elastase (IC_50_:0.032 μM) and α-chymotrypsin (IC_50_: 0.234 μM) from bovine pancreas [73]. Largamides are cyclic peptides isolated from *Lyngbyaconfervoides* and *Oscillatoria* sp. Largamides A-C inhibit elastase with IC_50_ ranges from 0.53 to 1.41 μM [74]. Chymotrypsin is also inhibited by argamides D through G, with IC_50_ values between 4 and 25 M [75]. Pompanopeptin A inhibits trypsin with IC_50_ of 2.4 μM [76]. Elastase and chymotrypsin were inhibited by Lyngbyastatin 4, a cyclic depsipeptide from Lyngbya sp., at 0.03 M [77]. With IC_50_s of 3.2–8.3 nM for elastase and 2.5–2.8 nM for chymotrypsin, ligbystatin 5–7 inhibit both enzymes [78]. Lyngbyastatin 8–10 inhibit elastase with IC_50_s of 120–210 nM [79]. Tiglicamides A-C and cyclodepsipeptides from the same source have IC_50_s that range from 2.14 to 7.28 M for inhibiting elastase [80]. The pitipeptolides A and B inhibit elastase activity at 50 μg mL^−1^ [81]. Somamide B from the same source inhibits elastase (9.5 nM) and chymotrypsin (4.2 µM) [78]. Cathepsins D and E are lysosomal proteases having anti-apoptotic functions and which play an important role in PCa [82,83]. Grassystatins A and B depsipeptides strongly inhibit cathepsins D (IC_50_:26.5 and 7.27 nM, respectively) and E (IC_50_:886 and 354 pM), whereas grassystatin C inhibits cathepsins D (IC_50_:1.62 µM) and E (IC_50_:42.9 nM) [84].

The cytoskeletal microfilament, actin, is required for cytokinesis, cell migration, and a host of other processes crucial for the stability of cancerous cells. Inhibiting actin polymerization slows the growth of metastatic neoplastic cells by causing the breakdown of microfilaments, which, in turn, reduces cell motility [85]. Jaspamide, a cyclicdepsipeptide from sponge (*Jaspis johnstoni)*, has shown antiproliferative activity against DU145, LNCaP, and PC-3 with IC_50_s of 0.8, 0.07 and 0.3µM by actin filament disruption. The same peptide has shown anticancer activity in a DU-145 xenograft [85].

Voltage-gated sodium channels (VGSC) are considered to have a role in cancer cell invasion and metastasis. In PCa, VGSC overexpression is crucial for cell movement and invasiveness [86,87,88]. Palmyramide A, a cyclic depsipeptide with an IC_50_ of 17.2 μM, and hermitamides A and B (lipopeptides) with IC_50s_ of 1 μM have been shown to block sodium channels via VGSC inhibition [89,90].

### 2.4. Antiangiogenic Effect

Angiogenesis is crucial in the development of cancer [90]. VEGF is produced in cancer cells and is required for angiogenesis. During low oxygen (hypoxia) periods, Mucin 1 (MUC1) increases HIF-1α to stimulate tumor development and angiogenesis. Its overexpression restricts apoptosis via upregulating Bcl-xL and inactivating BAD protein. A decline in VEGF expression level is associated with MUC1 silencing, establishing that MUC1 downregulation has an anti-angiogenic impact [91,92,93]. TFD and SIO peptides from fish inhibit PC-3 and DU145 cell migration by decreasing *VEGFR1* and *MUC1* protein expression [44,45,94].

### 2.5. Cell Cycle Arrest

Cell cycle arrest limits cell viability and is related to apoptosis [95]. The two main regulators of G2/M transition/progression are cdc2 and cell division cycle-25C (cdc25C). Multiple signaling pathways influence their regulation in the cell cycle, and are linked to carcinogenesis and tumor formation. cdc2 and cdc25C have been shown to enhance mitotic cell G2/M transition by dephosphorylating cyclin-dependent kinase-1 (CDK1) and activating the cyclin B1/CDK1 complex. Their downregulation causes G2/M cell cycle arrest via p53-mediated signal transduction [96]. cdc2 and cdc25C are highly expressed in PCa [97]. Chromopeptide A promotes G2/M phase arrest in PCa cells by suppressing cdc2 and cdc25C phosphorylation [51]. Similarly, cryptophycin-52 induces G2/M phase arrest in DU145 and LNCaP cells [24].

### 2.6. p53 Upregulation

The functional tumor protein p53 (p53) protein takes part in apoptosis initiation via BAK, BAX increment and Bcl2, Bcl-xL decrement. Cells also arrest in the G1 and G2/M stages when p53 is activated. Low p53 level has been detected in PCa [98,99,100]. Cryptophycin-52 [24], chromopeptide A [51] and sepia ink peptides [15,44,45] induce p53 upregulation in PC-3, DU145 and LNCaP cells and hence regulate p53-dependent apoptosis and cell cycle arrest.

### 2.7. Stimulation of Histone Hyperacetylation

Histone deacetylases (HDACs) are widely produced and over-activated in PCa. Stimulation of histone hyperacetylation in tumor through cellular HDAC inhibition results in G2/M phase arrest, apoptosis, activates p53, DNA-damage response and inhibition of metastasis and angiogenesis [101]. The chromopeptide A from marine-derived *Chromobacterium* sp. stimulates histone hyperacetylation by HDAC inhibition in PC-3, DU145, LNCaP cell lines and human PC-3 xenograft mouse model [51].

### 2.8. Mitochondrial Dysfunctions and Oxidative Damage

Reactive oxygen species (ROS) accumulation induces oxidative stress caused by mitochondrial abnormalities, and malignant cells require high ROS concentrations [102]. The most frequent type of DNA damage is DNA fragmentation, a direct consequence of oxidative stress [103]. Dolastatin 10 induces DNA damage in DU145 [19]. Similarly, Cryptophycin 52 and *C*-phycocyanin induce DNA damage in DU145 and LNCaP [24,48]. A schematic representation of the anticancer mechanisms of marine peptides is depicted in Figure 4 as under:

### 2.9. Unidentified Mechanisms for Anticancer Activity

Geodiamolides D–F [26], homophymines A–E [27], milnamides A–G [26], neamphamides B–D [30], rolloamide A [36], yaku’amides A and B [104] from sponges; lagunamide C [23], coibamide A [25], laxaphycin B [39] from cyanobacteria exhibit strong cytotoxicity in several PCa cells, although the specific targets are yet unknown. Patellamides B and F, ulithiacyclamide [38], trunkamide A [12,37], tamandarins A-B [32,33] from ascidia also elicit anti-PCa activity via unrevealed process. Microsporin A [17], sansalvamide A [34] and zygosporamide [35] from marine derived fungus; YALPAH from fish [105] possess anti-PCa properties via an unknown mechanism. Some of the anticancer effects of marine peptides are summarized in Table 1:

**Table 1 marinedrugs-20-00466-t001:** Summary of the sources, active peptides and anticancer mechanisms of action of Marine peptides.

Peptides	Marine Sources (Species Name)	Active Derivative	Anticancer Mechanisms	References
Aurilide B	Cyanobacteria (*Lyngbya majuscula*)	Cyclic depsipeptide	Microtubule stabilization	[22]
Lagunamide C			↓ cell viability	[23]
Cryptophycin-52 (LY355703)	Cyanobacteria (*Nostoc* sp.)		DNA fragmentation; Bcl2 ↓; Bax ↑; Bcl-xL ↓; caspase 3, 7 ↑; PARP ↑; p53 ↑; G2/M phase arrest; Microtubule depolymerization	[24]
Coibamide A	Cyanobacteria (*Leptolyngbya* sp.)		↓ cell viability	[25]
Laxaphycin B	Cyanobacteria (*Lyngbya majuscula*)	Cyclic dodecapeptide		[39]
*C*-phycocyanin	Cyanobacteria (*Limnothrix* sp.)	Peptide	Caspases 3, 9 ↑; cyt c release ↑; DNA fragmentation	[48]
Bisebromoamide	Cyanobacteria (*Lyngbya* sp.)		↓ cancer cell growth	[106]
Jaspamide	Sponge *(Jaspis johnstoni )*	Cyclic depsipeptide	Actin filament disruption	[28]
				[29]
Homophymines A–E	Sponge (*Homophymia* sp.)		↓ cell viability	[27]
Neamphamides B–D	Sponge (*Neamphius huxleyi*)			[30]
Geodiamolides D–F	Sponge (*Pipestela candelabra*)			[26]
Milnamides A–G		*N*-methylated linear peptide		
Rolloamide A	Sponge (*Eurypon laughlini*)	Cyclic heptapeptide		[36]
HTI-286	Sponge (*Hemiasterella minor*)	Tripeptide	Microtubule depolymerization	[14]
				[107]
Kahalalide F	Mollusk *(Elysia rufescens)*	Cyclic depsipeptide	PI3K-AKT inhibition; ErbB3 depletion	[31]
			↓ cancer cell growth	[108]
Elisidepsin			PI3K-AKT inhibition; ErbB3 depletion	[56]
Dolastatin 10	Mollusk *(Dolabella auricularia)*	Pentapeptide	Microtubule depolymerization	[19]
MCH	Mollusk (*Mytilus coruscus*)	Peptide	Bcl2 ↓; Bax ↑; caspase 3, 9 ↑	[43]
KLH	Mollusk (*Megathura crenulata*)		↓ cancer cell growth	[109]
Tamandarin A	Ascidia *(Trididemnum solidum)*	Cyclic depsipeptide	↓ cell viability	[32]
Tamandarin B				[33]
Patellamide B	Ascidia (*Lissoclinum patella*)	Cyclic octapeptide		[38]
Patellamide F				
Ulithiacyclamide		Cyclic peptide		
Trunkamide A	Ascidia *(Lissoclinum* sp.)	Cyclic heptapeptide		[12,37]
Diazonamide A	Ascidia *(Diazona angulata)*	Macrocyclic peptide	Microtubule depolymerization	[64]
Chromopeptide A	Bacteria (*Chromobacterium* sp. HS-13-94)	Depsipeptide	caspase 3 ↑; PARP cleavage; HDAC inhibition; G_2_/M phase arrest; p53 ↑	[51]
Sansalvamide A	Fungus (*Fusarium* sp.)	Cyclic depsipeptide	↓ cell viability	[34]
Microsporin A	Fungus (*Microsporum* cf. *gypseum*)	Cyclic tetrapeptide		[17]
Zygosporamide	Fungus *(Zygosporium masonii)*	Cyclic pentadepsipeptide	↓ cancer cell growth	[35]
SHP	Fish (*Sepia esculenta*)	Tripeptide	Bcl2 ↓; Bax ↑; caspase 3 ↑; p53 ↑	[15]
SIO			Bcl2 ↓; Bax ↑; caspase 3 ↑; p53 ↑; *VEGF* ↓	[44,45]
			S and G_2_/M phase cell cycle arrest	[16]
TFD	Fish (*Gadus* sp.)	Peptide	*VEGFR1* ↓; *MUC1* ↓	[94]
YALPAH	Fish *(Setipinna taty)*		↓ cancer cell growth	[105]
YALRAH				
YALPAR				
YALPAG				
ILYMP	Clam (*Cyclina sinensis*)	Pentapeptide	Bcl2 ↓; Bax ↑; caspase 3, 9 ↑; cyt c release ↑	[20]
SCH-P9 and SCH-P10	Clam (*Sinonovacula constricta*)	Tetrapeptide		[18]
AAP-H	Sea anemone (*Anthopleura anjunae*)	Oligopeptide		[21]
Tachyplesin	Horseshoe crab *(Tachypleus tridentatus)*	*Cyclic peptide*	caspase 3, 6, 7, 8, 9 ↑; cyt c release ↑	[50]

## 3. Clinical Trial Status

Several marine peptides with potential PCa efficacy are presently undergoing clinical trials. Soblidotin (TZT-1027) has shown efficacy in DU145 cell lines and has entered a phase I clinical trial (Table 2). It was designed to maintain significant anticancer activity while lowering the toxicity of the parent medication, dolastatin 10 [110,111,112]. Tasidotin/synthadotin (ILX651), a dolastatin 15 derivative is in Phase II clinical trial for hormone refractory PCa [113].

Didemnin B displays anti-PCa activity, and has progressed into phase II studies. Due to its high toxicity, low solubility, and short life span, clinical studies were halted favouring second generation dehydrodidemnine B, (aplidin or plitidepsin). Dehydrodidemnine B is in phase III clinical trials [10,32,114].

In phase I study, Kahalalide F was effective against PCa, with a favorable safety profile [115]. It was withdrawn from phase II due to lack of efficacy, short half-life, restricted range of activity, and poor patient response. However, given this compound’s potent cytotoxicity, it has facilitated the development of synthetic analogues to overcome its limitations by increasing its potency and half-life [63,116]. Elisidepsin (Irvalec^®^), one of PharmaMar’s most powerful Kahalalide F analogues, has progressed to phase II clinical trial due to its superior efficacy and nontoxic profile [117]. The preclinical studies (in vivo, in vitro) are separated according to xenograft’s approach and listed in Table 3.

**Table 2 marinedrugs-20-00466-t002:** Marine peptides as anticancer agents in clinical trials.

Cell lines/ (Peptides)	Phase	Clinical Trials.Gov Identifier	References
DU145 (Soblidotin)	Phase I	NCT00072228	[22,110,111]
Dolastatin 10	Phase II	NCT00003626	[23,110]
Tasidotin/synthadotin (ILX651)	Phase II	NCT00082134	[24,113]
Dehydrodidemnine B	Phase III	NCT00780975	[10,25,32]
Kahalalide F	Phase I	NCT00106418	[39,115]
Elisidepsin (Irvalec^®^)	Phase II	NCT00884845	[48,117]

## 4. Conclusions and Future Perspectives

Worldwide, PCa is a major cause of cancer-related mortality in men. While its frequency has increased, present therapy options are limited and have adverse effects, with relapses often occurring, highlighting the need for novel cancer treatments.

Information is scant on the use of marine peptides to treat this malignancy [5,6,118]. Marine peptides have been shown to have multiple anticancer effects. To date, most of the research has focused on the effects of marine peptides in vitro, making it difficult to extrapolate their in vivo efficacy. Only a few of these drugs have advanced to clinical trials. Although individual pharmacologically active marine peptides have been excluded from further drug discovery due to toxicity, there is a push to assess corresponding analogues for their efficacy. Substitution with D-amino acids, cyclization, pegylation, nanoparticles encapsulation, and XTEN conjugation can be used to overcome short half-life and metabolic instability. Namely, immunogenicity has been reduced by D-amino acid substitution [11,119,120,121]. Protein hydrolysates represent a rich source of antiproliferative, anticancer, and antioxidant compounds. Additional studies on the cell cycle phase arrest and increased apoptotic rates by these analogues are required to determine the pharmacological efficacies of protein hydrolysates.

The marine world offers a diversity of potential novel anticancer medications. However, there are several serious drawbacks to marine peptides, such as their stability in vivo. They are extremely vulnerable to cleavage by serum proteases in vivo, have a brief half-life, do have poor bioavailability, and provide manufacturing and production issues. The development of marine pharmaceuticals would benefit by interdisciplinary collaborations to overcome existing constraints. These innovative findings must be quickly translated into treatments for prostate cancers. Additionally, whether employed alone or in combination with a number of other chemotherapy agents, marine medicines and comparable generic compounds may provide insights into prospective clinical anticancer therapeutics. Additionally, to find out new concepts in the discovery of marine natural products in the future, analytical methods should be combined to the use of computational genetics, gene mining, experimental therapies, and other ground-breaking techniques.

Finally, further research into marine peptides and their mechanisms of action will be needed, resulting in an invaluable source of novel and potent new medications for the treatment of prostate cancer, and a better understanding of their mechanisms of action and their putative target sites. Taking into account the already available clinical trials’ data, the upcoming studies might utilize them for further clinical trials.

## Figures and Tables

**Figure 1 marinedrugs-20-00466-f001:**
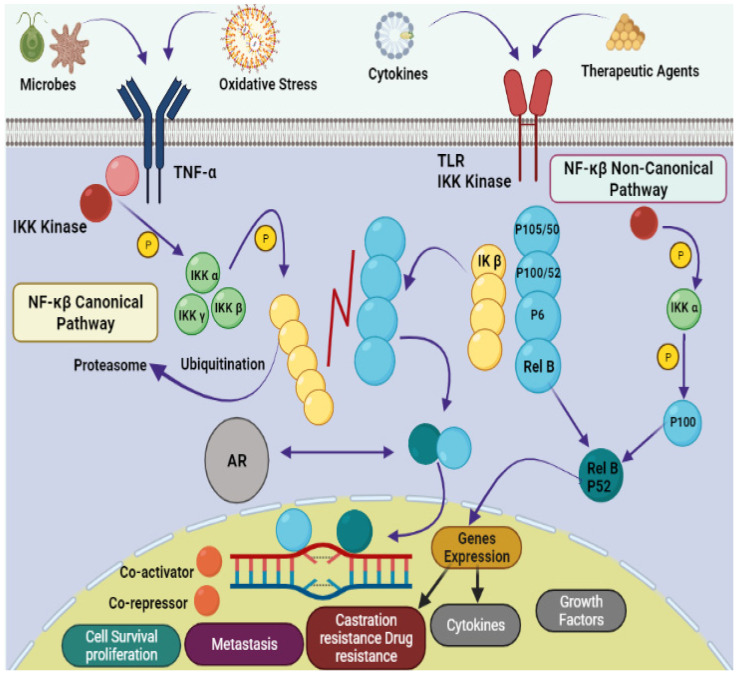
Pathophysiology of prostate cancer via NF-κB canonical and non-canonical pathways. The leading dimers of NF-κB are P50-P65, which activate the transcription process in canonical pathways. Different subunits like Toll-like receptor (TLR); tumor necrosis factor receptor (TNF-R); Inhibitor of NF-κB (IκB); IκB kinase; NF-κB-inducing kinase (NIK); mitogen-activated protein kinase (MAP); androgen receptor (AR); bone marrow-derived cell (BMDC) and major histocompatibility complex (MHC) are also involved in the pathology of prostate cancer.

**Figure 2 marinedrugs-20-00466-f002:**
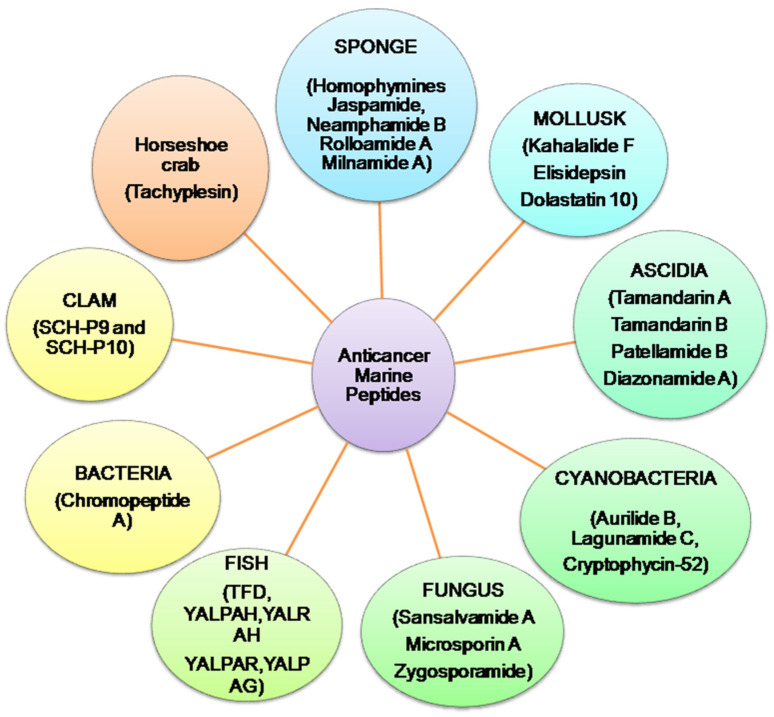
Summary of anticancer marine peptides isolated from different marine sources.

**Figure 3 marinedrugs-20-00466-f003:**
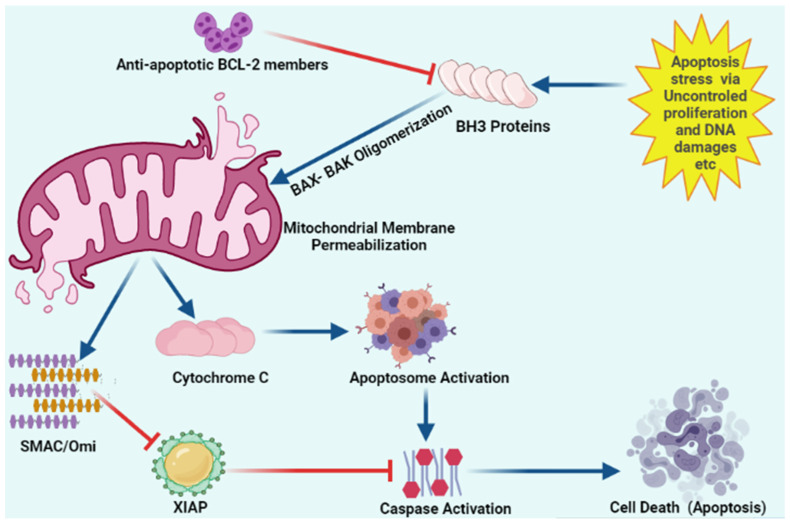
Schematic representation of intracellular apoptosis pathway.

**Figure 4 marinedrugs-20-00466-f004:**
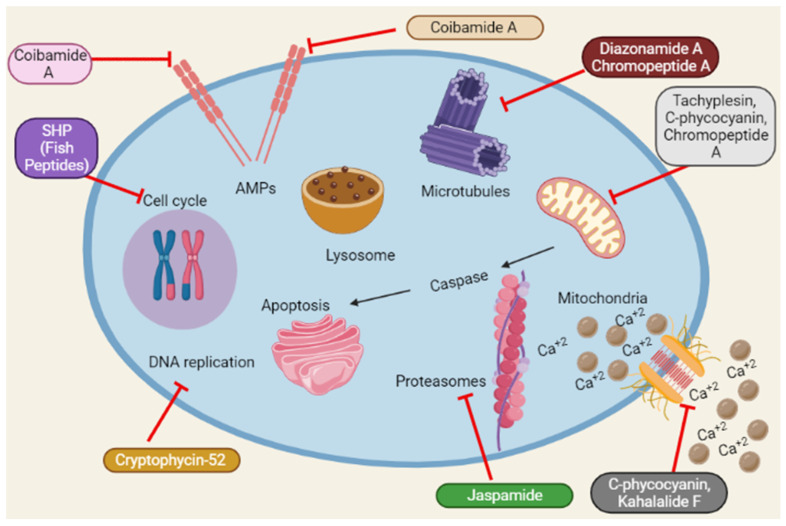
Summary of the schematic representation of the anticancer mechanisms of marine peptides at different cellular pathways. Marine peptides inhibit different pathways such as inhibition of cell cycle, Amps, caspase, Ca^+2^ influx, DNA replication, protein synthesis and lysosomal pathways.

**Table 3 marinedrugs-20-00466-t003:** Marine peptides as anticancer agents in pre-clinical trials.

In Vitro		In Vivo		References
Human Prostate Cancer Cell Lines	IC_50_	Experimental Model	Dose	
DU145 and PC-3 Aurilide B	<10 nM	-----	-----	[22]
PC-3 Lagunamide C	2.6 nM	-----	-----	[23]
DU145 and LNCaP Cryptophycin-52	1–10 pM	-----	-----	[24]
DU145 and PC-3 Coibamide A	300 ng mL^−1^	-----	-----	[10,25,32]
PC-3 Laxaphycin B	0.58 µM	-----	-----	[39]
LNCaP *C*-phycocyanin	500 µg mL^−1^	-----	-----	[48]
DU145 and PC-3 Bisebromoamide	GI50:40 nM	-----	-----	[106]
DU145 Jaspamide	0.8 µM	DU-145 xenograft	10 mg/kg s.c.	[28]
LNCaP Jaspamide	0.07 µM	-----	-----	
PC-3 Jaspamide	0.3 µM	-----	-----	
TSU-Pr1 Jaspamide	170 nM	-----	-----	[29]
PC-3 Homophymines A–E	A:4.2, B:6.2, C:3.0, D:6.3, E:3.9 nM	-----	-----	[27]
LNCaP Neamphamides B-D	B:230, C:190, D:110 nM	-----	-----	[30]
PC-3 Geodiamolides D-F	B: 170, C:110, D:130 nM	-----	-----	
	D:33.1, E:118, F:155 nM	-----	-----	[26]
DU145 Rolloamide A	0.85 µM	-----	-----	[36]
LNCaP Rolloamide A	0.8 µM	-----	-----	
PC-3 Rolloamide A	1.4 µM	-----	-----	
PC3MM2 Rolloamide A	4.7 µM	-----	-----	
LNCaP, C4-2, PC-3, PC-3dR HTI-286	0.65–4.6 nM	PC-3 and PC-3dR xenografts	1.5 mg/kg i.v.	[14]
-----	-----	PC3-MM2 xenograft	1.0 mg/kg i.v.	[107]
PC-3 Kahalalide F	0.07 µM	-----	-----	[31]
DU145 and LNCaP	0.28 µM	-----	-----	
-----	-----	PC-3 and DU145 xenografts	123 μg/kg i.v.	[108]
PC-3 Elisidepsin	1.80 µM	-----	-----	[56]
DU145	1.26 µM	-----	-----	
DU145 Dolastatin 10	0.5 nM	DU145 xenograft	5 µg q4d i.p.	[19]
PC-3 MCH	LC50:0.94 mg mL^−1^	-----	-----	[43]
DU145 Tamandarin A	GI_50_:12.5 μg	-----	-----	[109]
	1.36 ng mL^−1^	-----	-----	[32]
PC-3 Tamandarin B	1.4 µM	-----	-----	[33]
DU145 and PC-3 Patellamide B Patellamide F	LC_50_: 48 µM	-----	-----	[38]
	LC_50_: 13 µM	-----	-----	
	LC50: 3 µM	-----	-----	
DU145 Trunkamide A	7.08 nM	-----	-----	[12,37]
PC-3 Diazonamide A	2.3 nM	-----	-----	[64]
	2.43 nmol L^−1^	PC-3 xenograft	1.6 mg/kg i.v.	[51]
DU145 Chromopeptide A	2.08 nmol L^−1^	-----	-----	
LNCaP Chromopeptide A	1.75 nmol L^−1^	-----	-----	
PC-3 Sansalvamide A	27.4 μg mL^−1^	-----	-----	[34]
DU145 and PC-3 Microsporin A Zygosporamide	2.7 µM	-----	-----	[17]
	GI_50_:9.1 µM	-----	-----	[35]
PC-3 SHP	15 mg mL^−1^	-----	-----	[15]
DU145	1 mg mL^−1^	-----	-----	[44,45]
DU145 and PC-3 SIO	15 mg mL^−1^	-----	-----	[16]
PC-3 TFD, YALRAH, YALPAH, YALPAG, YALPAR	3.5 nM	-----	-----	[94]
	GI50:16.9 μM	-----	-----	[105]
	GI50:11.1 μM	-----	-----	
	GI50:19.0 μM	-----	-----	
	GI50:71.2 μM	-----	-----	
DU145, ILYMP, SCH-P9 and SCH-P10	11.25 mM	-----	-----	[20]
	SCH-P9:1.21, SCH-P10: 1.41 mg mL^−1^	-----	-----	[18]
PC-3	SCH-P9:1.09, SCH-P10: 0.91 mg mL^−1^	-----	-----	
DU145, AAP-H	2.298 mM	-----	-----	[21]
TSU, Tachyplesin	75 μg mL^−1^	-----	-----	[50]

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
