# Peer review of "Therapeutic Potential of Marine Peptides in Prostate Cancer: Mechanistic Insights"

_marinedrugs, 2022, doi:10.3390/md20080466_

Round 1
Reviewer 1 Report
The manuscript review entitled: "Therapeutic Potential of Marine Peptides in Prostate Cancer: 2 Mechanistic insights" is well written and presents a significant contribution to the field of natural products, specially marine-products, and deserves to be published. However, I have some notes that can better contribute to the manuscript, as follows:
In general text: Anticancer, not anti-cancer
A methodology strategy/section is lacking in the paper. It should be also included in the abstract. How was the criteria for selecting the articles? Which were the databases? Was there a period limit stablished (ex: papers published in the last decade)?
Table 1 is not mentioned in the text.
Lines 15-18 are in diffferent font size
Line 35: figure 1 are not consistent with globocan statistics. I believe it is not mentioned in the correct place.
Line 82: obtained from .... please insert a comma between mollusk and fish
Line 86: This review
Table 1 should be rearranged to fit all titles without cutting out the word. It should also include peptides that are in clinical phasis.
I suggest authors include a figure with the main mechanism of action described for the peptides. Figures 1 and 2 are good to understand the prostate cancer development and apoptosis pathways, respectively, but more interesting than these figures, is one reflecting the peptides actions on these well conceptualized pathways: apoptosis, cell cycle, angiogenesis…...
Please refer where the figures were created or adapted from...
Author Response
Reviewer #1
Comments and Suggestions for Authors
The manuscript review entitled: "Therapeutic Potential of Marine Peptides in Prostate Cancer: 2 Mechanistic insights" is well written and presents a significant contribution to the field of natural products, especially marine-products, and deserves to be published. However, I have some notes that can better contribute to the manuscript, as follows:
In general text: Anticancer, not anti-cancer
Reply: Changes have been done as per suggestion
A methodology strategy/section is lacking in the paper. It should be also included in the abstract. How was the criteria for selecting the articles? Which were the databases? Was there a period limit stablished (ex: papers published in the last decade)?
Reply: The literature review was done by utilizing different search engine database like PUBMED, GOOGLE SCHOLAR, SCIECNE DIRECT and SCOPUS etc. About 123 peer review and research articles were studied. There was no period of limits and till to date, updated literature review was done.
Table 1 is not mentioned in the text.
Reply: table 1 is mentioned and highlighted in the manuscript
Lines 15-18 are in diffferent font size
Reply: Font size is made uniform
Line 35: figure 1 are not consistent with globocan statistics. I believe it is not mentioned in the correct place.
Reply: the needful changes have been done and figure is recited and highlighted.
Line 82: obtained from .... please insert a comma between mollusk and fish
Reply: The needful changes have been done and highlighted
Line 86: This review
Reply: Changes done
Table 1 should be rearranged to fit all titles without cutting out the word. It should also include peptides that are in clinical phasis.
Reply: Table 1 is rearranged and for clinical trials table 2 is designed
I suggest authors include a figure with the main mechanism of action described for the peptides. Figures 1 and 2 are good to understand the prostate cancer development and apoptosis pathways, respectively, but more interesting than these figures, is one reflecting the peptides actions on these well conceptualized pathways: apoptosis, cell cycle, angiogenesis…...
Please refer where the figures were created or adapted from...
Reply: A separate figure (figure 4) is designed upon the valuable suggestions of the reviewer. Generalized ideas are taken from the literature review mentioned in the manuscript.
Regards
Prof. Dr. Haroon Khan
Reviewer 2 Report
The manuscript (marinedrugs-1805044) reviews the therapeutic Potential of Marine Peptides in Prostate Cancer through mechanistic insights. This review is of great significance to our understanding of marinepeptides in tumor research and provides many reflections for future research. The content is in line with the purpose of the journal. Therefore, I think that the manuscript can be acceptable for publication in Marine Drugs.
(1) It is suggested that the authors introduce the research of marine peptides on different tumors in the Introduction.
(2) Line 63-65: Can the authors analyze the distribution of anti-tumor peptides from different marine organisms in the form of figures
(3) Table 1: There are too many contents in this table. It is suggested that the authors divide it into several tables according to different species sources, which is more convenient for readers to read and more regular.
(4) The antitumor effects of Marine peptides were reviewed from different mechanisms. However, the mechanism of action of many peptides is multifaceted, and it is suggested that the authors can conduct a comprehensive analysis on sevel typical antitumor peptides.
(5) In future perspectives, it is suggested that the authors put forward constructive and feasibles suggestions for the future research according to the content of this review.
Author Response
Reviewer#2
Comments and Suggestions for Authors
The manuscript (marinedrugs-1805044) reviews the therapeutic Potential of Marine Peptides in Prostate Cancer through mechanistic insights. This review is of great significance to our understanding of marinepeptides in tumor research and provides many reflections for future research. The content is in line with the purpose of the journal. Therefore, I think that the manuscript can be acceptable for publication in Marine Drugs.
(1) It is suggested that the authors introduce the research of marine peptides on different tumors in the Introduction.
Reply: The updated needful data is incorporated in the introduction section and highlighted.
(2) Line 63-65: Can the authors analyze the distribution of anti-tumor peptides from different marine organisms in the form of figures
Reply: The needful suggested changes have been done and figure 2 is designed and highlighted
(3) Table 1: There are too many contents in this table. It is suggested that the authors divide it into several tables according to different species sources, which is more convenient for readers to read and more regular.
Reply: Table 1 is reorganized and table 2 is designed as per the suggestion of the reviewer
(4) The antitumor effects of Marine peptides were reviewed from different mechanisms. However, the mechanism of action of many peptides is multifaceted, and it is suggested that the authors can conduct a comprehensive analysis on sevel typical antitumor peptides.
Reply: The suggestions of the reviewer are highly appreciated as the review is not generalized to the antitumor effect of marine peptides but, a specified area of cancer i.e therapeutic effect of marine peptides on prostate cancer is focused in this review. Though different available marine peptides were specified and focused as therapeutic effects on prostate cancer but still peptides on different tumors were also discussed and highlighted
(5) In future perspectives, it is suggested that the authors put forward constructive and feasibles suggestions for the future research according to the content of this review.
Reply: The needful changes have been done and highlighted.
Regards
Prof. Dr. Haroon Khan